# Transcription Factor *MdbHLH093* Enhances Powdery Mildew Resistance by Promoting Salicylic Acid Signaling and Hydrogen Peroxide Accumulation

**DOI:** 10.3390/ijms24119390

**Published:** 2023-05-28

**Authors:** Hai Ma, Fuyan Zou, Dongmei Li, Ye Wan, Yiping Zhang, Zhengyang Zhao, Xiping Wang, Hua Gao

**Affiliations:** 1State Key Laboratory of Crop Stress Biology in Arid Areas, College of Horticulture, Northwest A&F University, Xianyang 712100, Chinazhaozy@nwsuaf.edu.cn (Z.Z.); 2Key Laboratory of Horticultural Plant Biology and Germplasm Innovation in Northwest China, Ministry of Agriculture, Northwest A&F University, Xianyang 712100, China

**Keywords:** apple, *MdbHLH093*, powdery mildew, disease resistance, salicylic acid signal, hydrogen peroxide

## Abstract

Powdery mildew is an apple disease caused by the obligate trophic fungus *Podosphaera leucotricha.* Basic helix–loop–helix (bHLH) transcription factors play important roles in plant development and stress responses, and they have been widely studied in model plants such as *Arabidopsis thaliana.* However, their role in the stress response of perennial fruit trees remains unclear. Here, we investigated the role of *MdbHLH093* in the powdery mildew of apples. The expression of *MdbHLH093* was significantly induced during the infection of apples with powdery mildew, and the allogenic overexpression of *MdbHLH093* in *A. thaliana* enhanced the resistance to powdery mildew by increasing the accumulation of hydrogen peroxide (H_2_O_2_) and activating the salicylic acid (SA) signaling pathway. The transient overexpression of *MdbHLH093* in apple leaves increased the resistance to powdery mildew. Conversely, when *MdbHLH093* expression was silenced, the sensitivity of apple leaves to powdery mildew was increased. The physical interaction between MdbHLH093 and MdMYB116 was demonstrated by yeast two-hybrid, bi-molecular fluorescence complementation, and split luciferase experiments. Collectively, these results indicate that MdbHLH093 interacts with MdMYB116 to improve apple resistance to powdery mildew by increasing the accumulation of H_2_O_2_ and activating the SA signaling pathway, as well as by providing a new candidate gene for resistance molecular breeding.

## 1. Introduction

During growth, development, and reproduction, plants inevitably suffer from various biotic and abiotic stresses [1]. Powdery mildew is an apple disease caused by the obligate trophic fungus *Podosphaera leucotricha* [2], which is widespread in all apple-producing areas of the world and causes immeasurable losses to farmers [3]. The fungus damages the different tissues and organs of the apple tree, including new shoots, leaves, flowers, and young fruits. The symptoms of powdery mildew first appear on leaves, and the disease is characterized by leaf curling, white powder production and deposition, dry leaf tips, and leaf shedding, which greatly affect the tree’s potential and fruit quality [4]. In current agricultural practices, powdery mildew in apples is mainly controlled through the use of bacteriostatic agents. Although the loss of apple yield can be controlled to a certain extent, long-term use is cost-prohibitive, adversely affects the environment [5], and fails to fundamentally address the cause of the disease. Therefore, it is important to identify and characterize the disease-resistance genes and to reveal their molecular mechanisms of action, which will help to improve the disease resistance of apples.

Plant life activities depend on the regulation of various transcription factors, among which the bHLH family of transcription factors is the second largest in plants [6]. These transcription factors are named for their 50 to 60-amino-acid bHLH domains, which consist of two highly conserved but distinct regions: namely, the N-terminal basic amino acid region and the C-terminal helix–loop–helix (HLH) [7]. Previous studies have reported that bHLH-family transcription factors play important roles in plant growth and development, as well as in resisting biotic and abiotic stresses. Numerous studies have demonstrated that the members of this transcription factor family are involved in the development of tissues and organs such as the carpel, anther, epidermal cells [8], stomata [9], and root [10]. Additionally, the bHLH-family transcription factors enhance plant resistance to abiotic stresses, and they can also promote tolerance to drought, high salt stress, low temperature, iron deficiency, heavy metal stress, and osmotic conditions [11,12]. Members of this transcription factor family are also important in the plant responses to various biotic stresses. For example, in *Arabidopsis thaliana*, the bHLH-family transcription factor *HBI1* is a negative regulator of the defense response, and the loss of *HBI1* increases resistance to bacterial infection, whereas overexpression of *HBI1* reduces the pathogen-induced immune response [13]. The transient overexpression of *StCHL1* significantly increases the ability of *Phytophthora* to infect tobacco leaves, which is consistent with the finding that its congeners *HBI1* and *CIB1* are negative regulators of the immune response [14]. The overexpression of *TabHLH060* in *A. thaliana* significantly increases sensitivity to *Pseudomonas clove* DC3000 and decreases the transcriptional levels of the resistant genes *PR1*, *PR2*, and *PR5* that are involved in the salicylate (SA) signaling pathway; whereas, the transcriptional levels of the *PDF1.2* and *ORA59* involved in jasmonic acid and ethylene signaling pathways are upregulated, indicating *TabHLH060* negatively regulates plant disease resistance through jasmonic acid and ethylene signaling pathways [15]. In soybean, *GmPIB1* inhibits the expression of *SPOD1*, a key enzyme in the production of reactive oxygen species—which can reduce their production and improve soybean resistance to phytophthora disease [16].

However, the regulatory mechanisms of the bHLH family of transcription factors that are involved in powdery mildew are poorly understood. A study performed in wheat showed that the expression levels of 28 *TabHLH* genes and 6 *TabHLH* genes were significantly downregulated and upregulated in powdery mildew, respectively [17]. The heterologous expression of the pumpkin bHLH-family transcription factor *CmbHLH87* in tobacco plants alleviates powdery mildew symptoms in leaves, accelerates cell necrosis, and enhances H_2_O_2_ accumulation. The expression levels of *PR1a*, *PR5*, and *NPR1* in transgenic plants were also significantly higher than those in control plants [18]. A differentially expressed gene was screened from the transcriptomic data of apple inoculated with *P. leucotricha*, and according to the study of Yang et al., its homologous gene in *A. thaliana* was *AtbHLH093* [19], so the gene was named *MdbHLH093*. When apple leaves were infected with *P. leucotricha*, the expression level of *MdbHLH093* was significantly increased [20]. We hypothesized that *MdbHLH093* may have a certain regulatory effect on powdery mildew infection in apples. The *Arabidopsis* bHLH-family transcription factor *AtbHLH93* was observed to promote flowering in plants under short-day conditions by regulating the expression of genes associated with erythrominin biosynthesis and metabolism [21]. The knockdown of *bHLH93* in tobacco significantly weakens the allergic response induced by the bacterial wilt effector protein RipI and reduces the expression of the defense gene *PDF1.2* [22]. For the transcription factor *MdbHLH093*, a previous study demonstrated that it directly activates the transcription of *MdSAG18*, promotes the expression of age-related genes, and regulates the plant-aging process [23]. To clarify the relationship between *MdbHLH093* expression and powdery mildew in apples, we performed functional analyses through the allogenic transformation of *A. thaliana* and the transient transformation of apple leaves. In subsequent experiments, we observed that MdbHLH093 physically interacts with MdMYB116 and plays an active role in the SA-mediated powdery mildew defense. The aim of this study is to identify the candidate genes for breeding apple varieties resistant to powdery mildew and to provide a foundation for studying the molecular mechanism of apple resistance to powdery mildew.

## 2. Results

### 2.1. Sequence Analysis and Evolutionary Tree Construction of MdbHLH093

In the apple genome, *MdbHLH093* is located on chromosome 15 (https://www.rosaceae.org, accessed on 5 March 2022). The coding sequence (CDS) of *MdbHLH093* is 1095 bp in length, which encodes a 364-amino acid (aa) protein. The MdbHLH093 protein contains a highly conserved HLH domain (198–247 aa) (Figure 1a). The phylogenetic tree of MdbHLH093 and its homologous proteins in 10 species—namely, *A. thaliana*, rice, wheat, corn, tobacco, tomato, grape, pear, cherry, and peach—was constructed. The MdbHLH093 in apple was closely related to the homologous genes in pear (Figure 1b). The amino acid sequences of these genes were compared using DNAMAN 6.0 software, all genes were contained in an HLH domain and were highly conserved (Figure 1c).

### 2.2. Expression Pattern Analysis of MdbHLH093

To evaluate the potential regulatory effect of the *MdbHLH093* transcription factor on apple powdery mildew, we analyzed the expression pattern of *MdbHLH093* in apple leaves inoculated with powdery mildew. The *MdbHLH093* level was upregulated after the inoculation of apples with powdery mildew. However, it was significantly lower than that of the control group at 12 h post-inoculation (hpi) and 24 hpi, whereas after 48 hpi, the expression level was significantly upregulated, peaking at 72 hpi, and it was significantly increased at 48 hpi, 72 hpi, and 96 hpi when compared to the controls (Figure 2a). To analyze the spatial regulation where *MdbHLH093* may play a major role, we analyzed the expression of *MdbHLH093* in roots, stems, leaves, flowers, and fruits. *MdbHLH093* was expressed in all tissues, but mainly in leaves, and its level in leaves was more than 100-fold higher that of the other tissues (Figure 2b), indicating *MdbHLH093* mainly functions in leaves.

To determine the specific location of the MdbHLH093 protein in cells, the open reading frame (ORF) of *MdbHLH093* was cloned downstream of the cauliflower mosaic virus (CaMV) 35S promoter with green fluorescent protein (GFP), and the plasmid was heterogeneously expressed in tobacco. Confocal microscopy was used to detect the GFP in the nucleus, indicating MdbHLH093 was localized in this ultrastructure (Figure 2c). To determine whether the transcription factor had transcriptional activation activity, we introduced the pGBKT7 vector containing the *MdbHLH093* gene into yeast receptor cells by the yeast two-hybrid method and observed that it could reduce tryptophan deficiency. The cells inoculated in the medium containing X-α-Gal and aureobasidin A (SD/-Trp/X-α-Gal/AbA) grew normally and the medium turned blue, indicating *MdbHLH093* had transcriptional activation activity. To confirm the presence of the transcriptional activation region and to avoid self-activation in subsequent experiments, we truncated MdbHLH093 by intercepting 83, 117, and 167 aa at the C-terminus, and 143 and 197 aa at the N-terminus, respectively. The truncated fragments were fused into the pGBKT7 vector and transformed into yeast receptor cells. The truncated fragments at the C-terminus showed transcriptional activation activity, whereas the truncated fragments at the N-terminus eliminated self-activation (Figure 2d), indicating the transcriptional activation region may be located within the N-terminus (1–146 aa). Thus, *MdbHLH093*^△1−146^ was selected for follow-up studies.

### 2.3. Heterologous Expression of MdbHLH093 in A. thaliana Enhances Resistance to Powdery Mildew

To determine whether *MdbHLH093* has a regulatory effect on apple resistance to powdery mildew, we generated transgenic *A. thaliana* plants expressing *MdbHLH093* under the control of a strong CaMV 35S promoter (35S: *MdbHLH093*) and selected three transgenic lines with significantly upregulated expression levels to evaluate the apple resistance to powdery mildew (Figure 3b). After 7 days of infection, compared to wild type (WT) plants, the three transgenic lines had a significantly lower incidence of infection, fewer infected leaves, and no leaf chlorosis (Figure 3a). The number of spores on the leaves of WT and transgenic *A. thaliana* plants was counted, and the number of spores on the leaves of transgenic lines was significantly lower than those on the wild type leaves (Figure 3c). Additionally, staining with trypan blue, diaminoaniline (DAB), and aniline blue was performed on WT and transgenic *Arabidopsis* leaves to observe the lesion area, as well as H_2_O_2_ and callose accumulation. From trypan blue staining, the WT plants had larger lesions and a more severe disease. From DAB staining, the H_2_O_2_ accumulation in transgenic *A. thaliana* plants was higher than those in WT plants, which may be one of the mechanisms of enhanced disease resistance. Aniline blue staining showed no significant difference between the WT and transgenic lines, indicating *MdbHLH093* overexpression does not affect the accumulation of callose to regulate disease resistance (Figure 3d). In addition, *A. thaliana* leaves infected with powdery mildew were stained with trypan blue at 0 hpi, 24 hpi, 72 hpi, and 120 hpi to observe pathogenic growth and disease development. Compared to WT plants, the germination of spores was inhibited in transgenic *A. thaliana* plants, as well as the growth and elongation of mycelia and the generation of secondary spores to a certain extent (Figure 3e); this indicates that *MdbHLH093* overexpression enhances the resistance of plants to powdery mildew.

### 2.4. MdbHLH093-Overexpressing Plants Increase the Expression of Genes Related to the SA Signaling Pathway

The SA-regulated defense response plays an important role in defending plants against eutrophic pathogens. Therefore, we analyzed the transcriptional levels of SA synthesis-related genes (*AtEDS1*, *AtICS1*) and SA signaling pathway-related genes (*AtPR2*, *AtPR5*). In transgenic plants, *AtEDS1* was significantly increased at 24 hpi, and its expression level remained elevated, whereas *AtICS1* expression increased gradually in both WT and transgenic plants, and the expression levels of *AtEDS1* and *AtICS1* were significantly higher than those in WT plants at 24 hpi, 72 hpi, and 120 hpi. Furthermore, *AtPR2*, a downstream gene of the SA signaling pathway, was significantly upregulated in powdery mildew. Specifically, the response was more intense after 72 hpi, the transcriptional level was significantly increased, and the extent of the increase was higher in transgenic lines than that in WT plants. *AtPR5* was also upregulated in powdery mildew, but the extent of the increase was lower than that of *AtPR2*. There was no significant difference compared to inoculated control plants at 24 hpi, and the expression level in transgenic lines at 72 hpi and 120 hpi was significantly higher than that in WT plants. Collectively, these results indicate *MdbHLH093* positively regulates apple resistance to powdery mildew by promoting SA biosynthesis and signaling pathways (Figure 4).

### 2.5. Transient MdbHLH093 Overexpression Enhances Apple Leaf Resistance to Powdery Mildew

To confirm the regulatory effect of *MdbHLH093* in powdery mildew in apples, we transiently transformed the overexpression vector containing *MdbHLH093* into Gala-3 apple tissue culture seedlings and inoculated *P. leucotricha* 24 hpi after penetration. We analyzed the *MdbHLH093* level at 0 hpi, 24 hpi, 72 hpi, 120 hpi, and 168 hpi after osmosis. The expression level increased and then decreased, peaking at 24 hpi, and the expression level was more than 2-fold higher than that in seedlings transformed with the empty vector (OE-EV). Furthermore, the transcription level at 24 hpi, 72 hpi, and 120 hpi was significantly increased compared to the controls, indicating *MdbHLH093* was successfully overexpressed (Figure 5b). The disease degree of the transient overexpression of *MdbHLH093* in inoculated plants was significantly less severe than that in WT plants and osmotic no-load controls (Figure 5a). Quantitative analysis of the number of spores revealed that the overexpressed strains had fewer spores (Figure 5c). Trypan blue staining was performed on apple leaves at 0 hpi, 48 hpi, and 144 hpi after inoculation, and the growth of *P. leucotricha* was inhibited, especially in terms of the production of secondary spores (Figure 5d). Additionally, we quantitatively analyzed the transcriptional levels of the key SA pathway genes *MdEDS1*, *MdICS1*, *MdPR2*, and *MdPR5* by real-time fluorescence. Upon *MdbHLH093* overexpression, the levels of both *MdEDS1* and *MdICS1* peaked at 48 hpi and increased gradually after 96 hpi, in which the *MdEDS1* level at 48 hpi and 96 hpi was significantly higher than that in the controls, and the *MdICS1* level at 48 hpi and 144 hpi was significantly higher than that in the controls. PR2, a protein related to the course of powdery mildew, was significantly higher in overexpressing plants than that in the controls at 48 hpi, 96 hpi, and 144 hpi. *MdPR5* showed an increasing and then a decreasing trend during infection, and its expression level was higher after 96 hpi, with significant differences at 96 hpi and 144 hpi (Figure 5e), indicating that *MdbHLH093* overexpression enhances apple resistance to powdery mildew through the SA signaling pathway.

### 2.6. Transient Silencing of MdbHLH093 Reduces Apple Resistance to Powdery Mildew

We constructed the *MdbHLH093* RNAi vector, transiently transformed it into apple plants, and examined the *MdbHLH093* expression level. The results showed that, compared with WT, the expression level of *MdbHLH093* was obviously increased in plants that were transformed into empty vectors (RNAi-EV); we speculated that this might be caused by vacuum stress. Compared with RNAi-EV, the expression of *MdbHLH093* in the experimental group (RNAi-*MdbHLH093*) was reduced by approximately two-thirds at 24 hpi, and by approximately one-third at 72 hpi and 120 hpi, indicating successful *MdbHLH093* knockdown (Figure 6b). Furthermore, the knockdown of *MdbHLH093* increased the incidence of *P. leucotricha* infection at 168 hpi in apple leaves (Figure 6a). Quantitative analysis showed that *MdbHLH093* knockdown significantly increased the number of spores in leaves (Figure 6c) and the sensitivity of plants to powdery mildew. Trypan blue staining also indicated that *P. leucotricha* growth on the leaves of *MdbHLH093*-silenced plants was enhanced, and the production of secondary spores was higher than that in the controls (Figure 6d). Additionally, we examined the transcriptional levels of key genes of the SA pathway; the results showed that after *MdbHLH093* knockdown, *MdEDS1* expression was significantly inhibited, its expression at 48 hpi was one-third that of the controls, and its expression at 96 hpi was reduced by approximately 50%. The expression levels of *MdICS1* and *MdPR2* at 48 hpi and 96 hpi were significantly decreased compared to the controls, whereas *MdPR5* expression was significantly decreased at 48 hpi, 96 hpi, and 144 hpi compared to the controls (Figure 6e). This indicates that the inhibition of the SA signaling pathway after *MdbHLH093* knockdown reduces disease resistance.

### 2.7. MdbHLH093 Interacts with MdMYB116

In our previous studies, we identified the *MdMYB116* transcription factor as a positive regulator of powdery mildew infection in apples and observed that it could enhance the resistance of apple to powdery mildew by promoting reactive oxygen species accumulation and SA signaling [24]. Other studies have reported that bHLH and MYB families can form dimers, thereby functioning in a synergistic manner [25]; thus, we hypothesized that MdbHLH093 and MdMYB116 may also be involved in a similar interaction. We used the yeast two-hybrid method to verify the presence of protein-protein interactions. Yeast colonies containing pGBKT7-*MdbHLH093*^△1−146^ and pGADT7-*MdMYB116* grew in a SD/−Ade−His−Leu−Trp medium. In the presence of X-α-Gal and AbA, the colonies grew normally and appeared blue; this was similar to the positive control (Figure 7a), indicating MdbHLH093 specifically interacts with the MdMYB116 in yeast cells. To verify this interaction in plants, we conducted bimolecular fluorescence complementation experiments (Figure 7b). Compared to the control, only NE-*MdbHLH093*+CE-*MdMYB116* produced a fluorescent signal in the nucleus. Split-luciferase assay was performed on tobacco leaves, and luciferase activity was detected at the site of co-injection of Nluc-*MdMYB116* and Cluc-*MdbHLH093*, whereas no luciferase activity was detected in the control leaves (Figure 7c), indicating MdbHLH093 interacts and co-localizes with MdMYB116 in the nucleus.

## 3. Discussion

Apple, one of the most popular edible fruits worldwide, has important economic value, but its production often suffers from various biotic stresses, such as powdery mildew. At present, powdery mildew in apples is mainly prevented through pesticide use, which not only increases costs, but also creates food safety and environmental pollution issues [26]. A better sustainable solution is to enhance the resistance of apples to powdery mildew in order to control the disease and to breed resistant varieties by exploring genes with specific resistance to powdery mildew or broad-spectrum resistance to fungal diseases. It was observed that NPR1, a major regulatory factor of the SA-mediated disease resistance pathway, enhances broad-spectrum resistance to diseases by altering the expression levels of disease-course-related proteins in plants [27,28]. Additionally, *MhNPR1* is overexpressed in the Fuji apple, where it regulates the expression of *MdPR* and *MdMLO* and improves the resistance to powdery mildew [29]. Thus, the loss of *MLO* can improve plant resistance to powdery mildew, which has been demonstrated in barley, *Arabidopsis*, tomato, pea, wheat, and cucumber [30], as well as confirmed in apple. It was observed that the knockout of *MdMLO19* significantly improves the resistance of apples to powdery mildew [5]. Previous studies have reported that some bHLH-family transcription factors control disease resistance through various pleiotropic mechanisms. For example, in tomato, the overexpression of *bHLH132*, a transcription factor, greatly improves resistance to *Xanthomonas euvesicatoria* [31]. In rice, the bHLH-family transcription factor *RERJ1* is involved in the jasmonate-mediated expression of stress response genes through a mechanism that involves its interaction with OsMYC2, which improves plant defense against herbivores and bacterial infection [32]. At present, most studies of biotic stress induced by bHLH-family transcription factors focus on model plants and bacterial diseases; as such, there are only a few studies on powdery mildew in apples.

In this study, the qPCR analysis showed that the *MdbHLH093* level was significantly decreased at 12 hpi and 24 hpi compared to the controls, and significantly increased at 48 hpi, 72 hpi, and 96 hpi, indicating *MdbHLH093* can be induced by *P. leucotricha* (Figure 2a). In addition, *MdbHLH093* overexpression in *A. thaliana* and apple leaves enhanced resistance to powdery mildew (Figure 3, Figure 4 and Figure 5), whereas *MdbHLH093* knockdown in apple leaves increased sensitivity to powdery mildew (Figure 6), indicating *MdbHLH093* has a positive regulatory effect on the resistance of apple to powdery mildew. Hydrogen peroxide plays an important role in plant resistance to stress. It kills pathogenic microorganisms, inhibits the germination of pathogenic spores and the production of plant protectants and antibacterial proteins, as well as induces the hypersensitivity of cell death system-acquired resistance [33]. Additionally, it regulates the secondary growth of cell wall structures, facilitates the lignification of cell walls after cell injury, and upregulates the expression of disease-resistance-related genes [34,35]. The overexpression of *MdbHLH093* in *A. thaliana* visibly increased the H_2_O_2_ content compared to the controls (Figure 3d), which may be one of the mechanisms involved in disease resistance. Callose is rapidly deposited between plasma membranes and cell walls after plants are attacked by disease agents, where it induces a specific defense mechanism. Thus, callose is an important index for evaluating the resistance of plants to disease [36]. Cui et al. demonstrated that the transient silencing of *VvCSN5* in grape leaves increases callose deposition, and thus induces resistance to powdery mildew [37]. However, in our study, no obvious change in the callose content was observed after *MdbHLH093* overexpression, indicating *MdbHLH093* does not increase the callose content in disease resistance (Figure 3d).

Plant hormones are important regulatory factors required for plant defense, and the complex signaling network composed of hormones enables plants to activate appropriate and effective defense responses against pathogens [38]. Salicylate plays an important role in the resistance of plants to eutrophic and semi-eutrophic pathogens [39], which allows plants to resist powdery mildew. A previous study has shown that *VqWRKY31* improves grape resistance to powdery mildew by activating SA biosynthesis and SA-related immune signaling pathways [40]. The wheat bHLH-family transcription factor *TabHLH60* negatively regulates the resistance of plants to disease by inhibiting the transcriptional levels of resistant genes *PR1*, *PR2*, and *PR5* in the SA signaling pathway [15]. Furthermore, the *CmbHLH87* overexpression in pumpkins increases the accumulation of H_2_O_2_, upregulates the transcriptional levels of *PR1a*, *PR5*, and *NPR1*, as well as enhances resistance to powdery mildew [18], indicating bHLH-family transcription factors regulate disease resistance by regulating hormonal pathways. Upon *MdbHLH093* overexpression in *A. thaliana*, the transcriptional levels of four SA signaling-related defense genes were upregulated compared to the controls (Figure 4). *AtEDS1* and *AtICS1* are the key genes of the SA biosynthesis pathway [41], whereas *AtPR2* and *AtPR5* are the key defense genes induced by SA [42]. In our study, the transient overexpression of *MdbHLH093* in apple leaves upregulated the transcriptional levels of four SA signaling-related defense genes, which is consistent with the study of *A. thaliana*. Whereas the knockdown of *MdbHLH093* downregulated the transcriptional levels of these genes, indicating *MdbHLH093* overexpression enhances resistance to powdery mildew through the SA signaling pathway.

The bHLH proteins co-regulate plant disease resistance by interacting with other proteins. For example, the rice bHLH-family transcription factor *RERJ1* interacts with members of the OsMYC2 and JAZ families to enhance the resistance of plants to insects and pathogenic bacteria through the regulation of the jasmonate pathway [32]. Apple MdbHLH92 interacts with MdERF100 to regulate the resistance of apple to powdery mildew [4]. In our previous study, *MdMYB116* increased H_2_O_2_ accumulation and SA signaling-related gene expressions to regulate the resistance to powdery mildew [24]. Furthermore, members of the bHLH family can interact and co-function with those of the MYB family. For example, the bHLH transcription factor *MYC5* interacts with MYB transcription factors *MYB21* and *MYB24* to form the bHLH–MYB transcription complex, which regulates stamen development [43]. In our study, the interaction between MdbHLH093 and MdMYB116 was verified by yeast two-hybrid, bi-molecular fluorescence complementation, and split luciferase experiments (Figure 7), and it was observed that this interaction had a synergistic effect on the positive regulation of powdery mildew resistance.

In conclusion, we found that *MdbHLH093* enhances resistance to powdery mildew by regulating H_2_O_2_ accumulation and SA signaling, and it interacts with the resistance gene *MdMYB116* to elicit synergistic effects (Figure 8). Further studies are needed to explore whether there is an association or a regulatory mechanism between H_2_O_2_ production and SA signaling. Our results provide new options for breeding disease-resistant plants and offer new insights into the functions and mechanisms of *bHLH* in response to pathogen inoculation.

## 4. Materials and Methods

### 4.1. Plant Materials and Growth Conditions

Apple (Gala-3) tissue culture seedlings rooted for 45 days were used, and the rooting medium was composed of 2.22 g/L Murashige and Skoog (MS), 20 g/L of sucrose, 500 μL of Indole-3-acetic Acid (IAA, 1 mg/mL), 500 μL of Indole-3-butyric Acid (IBA, 1 mg/mL), and 7.5 g/L of agar, pH 6.1. Seedlings were cultivated at a temperature of 25 °C and a photoperiod of 16 h light/8 h dark. *Arabidopsis* Columbia ecotype (Col-0) plants were grown in an incubator with a 16 h/8 h photoperiod and 70% relative humidity. *Nicotiana benthamiana* (line NC89) was grown in a chamber at 25 °C. Apple root, stem, leaf, flower, and fruit tissues were acquired from the White Water Apple Test Station of Northwest A&F University, Shaanxi, China.

### 4.2. Pathogen Inoculation

A 100 mL spore suspension (1 × 10^6^ spores/mL) was prepared using material collected from the farm of Northwest A&F University and was evenly sprayed on the leaves of rooting seedlings that were previously transplanted into the substrate for one month. Water served as the control. Temperature and humidity were maintained at 25 °C and 70%, respectively, and the seedlings were exposed to 16 h of light and 8 h of dark. Leaves were sampled at 0 hpi, 6 hpi, 12 hpi, 24 hpi, 48 hpi, 72 hpi, and 96 hpi for expression analysis.

### 4.3. Bioinformatics Analysis

Chromosome positions of *MdbHLH093* were predicted using Blast-Search in the Apple Genome Browser, and the conserved protein domains were predicted using the simple modular architecture research tool (SMART; http://smart.embl-heidelberg.de/, accessed on 5 March 2022). Ten homologous genes of *MdbHLH093* from NCBI (https://www.ncbi.nlm.nih.gov/, accessed on 5 March 2022) were downloaded, and the multi-sequence alignment of *MdbHLH093* was performed using DNAMAN 6.0 software. A phylogenetic tree of the two genes and their homologous genes was generated using the neighbor joining method, and 1000 bootstrap replicates were performed by MEGA 5.0 software.

### 4.4. Subcellular Localization Analysis

The full-length CDS of *MdbHLH093* (without stop codons) was cloned from Gala-3 apple cDNA by using primers designed by Primer Premier 5.0 software (Appendix A). In addition, the cloned fragment was inserted into the pCAMBIA2300-GFP recombinant vector. The *MdbHLH093*-GFP plasmid was transformed into *Agrobacterium tumefaciens* Gv3101 and penetrated into tobacco leaves [44]. Tobacco plants were grown for 2 days at a temperature of 25 °C, a relative humidity of 60%, and a photoperiod of 16 h light/8 h dark. Green fluorescent protein was observed with a laser scanning confocal microscope (model FV1000MPE; Olympus, Tokyo, Japan).

### 4.5. Arabidopsis Transformation and Disease Analysis

The pCAMBIA2300-*MdbHLH093*-GFP vector was heterogenetically transformed into *A. thaliana* by the inflorescence infection method [45]. T0 plant seeds were selected on an MS medium supplemented with 75 mg/L of kanamycin. Homozygous T3 strains were selected from 3 independent T1 strains for subsequent experiments. After the inoculation of wild type and transgenic strains with *Golovinomyces cichoracearum* UCSC1, the leaves were collected at 0 hpi, 24 hpi, 72 hpi, and 120 hpi for qPCR, and the infected leaves were stained with trypan blue to observe the growth of pathogenic bacteria. In brief, *A. thaliana* leaves were immersed in boiling trypan blue solution for 4 min and rinsed with water 2–3 times. Leaves were decolorized with 2.5 g/mL of chloral hydrate for 4–6 h and stored in 50% glycerol. At 168 hpi, the number of spores on leaves was counted by the blood cell counting plate method; the infected *A. thaliana* leaves were collected, added to 5 mL of sterile water, and mixed for 1 min until the spores were completely released. The number of spores was counted using a hematocytometer, and the leaves were stained with trypan blue, DAB, and aniline blue. For the DAB staining, the leaves were immersed in 1 mg/mL of DAB (pH 3.8) for 8 h, and then soaked in 95% ethanol to observe the accumulation of H_2_O_2_. Aniline blue staining was used to assess the deposition of the callose. In brief, leaves were decolorized with 95% ethanol, stained with aniline blue for 24 h, and observed with a fluorescence microscope (model BX63; Olympus).

### 4.6. Instantaneous Transformation of Apple Leaves Mediated by Agrobacterium tumefaciens and Inoculation with Powdery Mildew

Overexpression and RNA interference (RNAi) vectors (pK7GWIWG2D-*MdbHLH093*) were transformed into *Agrobacterium* Gv3101, cultured in liquid Luria-Bertani medium at 28 °C for 16 h in a 180 rpm shaking incubator, and centrifuged at 5000 rpm for 10 min. Bacteria were suspended in an infiltrating buffer (10 mM MES, 10 mM MgCl_2_, 200 μM acetyl syringone, pH 5.6) to OD_600_ = 0.8. Apple (Gala-3) tissue culture seedlings, which had been rooted for 45 days, were washed free of the root medium, placed in the resuspension solution upside down, soaked under vacuum conditions of 0.08 MPa for 15–20 min until the backs of leaves were completely water soaked, and then transplanted into substrate for growth at a temperature of 25 °C and a relative humidity of 90%. Each treatment consisted of at least 10 seedlings.

After vacuum osmosis for 24 h, the *P. leucotricha* were inoculated and maintained at a temperature of 25 °C, a relative humidity of 90%, and a photoperiod of 16 h light/8 h dark. Leaves were collected at 0 hpi, 48 hpi, 96 hpi, and 144 hpi for qPCR. Leaves inoculated at 0 hpi, 48 hpi, and 144 hpi were stained with trypan blue to observe the growth of the pathogenic bacteria. In brief, leaves were immersed in a boiling trypan blue solution for 10 min, soaked in trypan blue solution at room temperature for 15–20 h, rinsed with water 2–3 times, decolorized with 2.5 g/mL of chloral hydrate for 24 h, and then observed with a fluorescence microscope. Three biological replicates were performed to determine the number of spores in the infected leaves.

### 4.7. Yeast Two-Hybrid Assay

For the yeast transcriptional activation test, the full-length CDS of *MdbHLH093* and its truncated fragments were inserted into the pGBKT7 vector, and the plasmids were transformed into Y2H receptor cells, which were cultured in media (SD/Trp, SD/Trp + 200 ng/mL AbA, SD/Trp + 200 ng/mL AbA, and 50 μg/mL X-α-Gal) at 30 °C for 2–3 days. Colony growth and color changes were observed. The pGBKT7-53 combined with pGADT7 served as a positive control, whereas pGBKT7-Lam combined with pGADT7 or empty pGBKT7 served as a negative control.

To explore the interaction in yeast, the full-length CDS of *MdMYB116* was inserted into the pGADT7 vector and transformed into Y2H receptor cells together with the pGBKT7-*MdbHLH093*^△1−146^ vector. Cells were cultured in media (SD/−Leu−Trp [DDO], SD/−Ade−His−Leu−Trp [QDO], and SD/−Ade−His−Leu−Trp [QDO] + 200 ng/mL AbA + 50 μg/mL X-α-Gal) at 30 °C for 2–3 days. Colony growth and color changes were observed. Blue colonies indicated an interaction between the two proteins.

### 4.8. Bimolecular Fluorescence Complementary Analysis

The full-length CDS of *MdbHLH093* and *MdMYB116* (without their respective stop codons) were cloned into pSPYNE and pSPYCE vectors, respectively, and the plasmids were transformed into *Agrobacterium* Gv3101. pSPYNE-*MdbHLH093*/pSPYCE-*MdMYB116* was expressed instantaneously in the tobacco leaves by *Agrobacterium*-mediated transformation. pSPYNE+pSPYCE, pSPYNE-*MdbHLH093*+pSPYCE and pSPYNE+pSPYCE-*MdMYB116* served as the controls. After 48 h of conversion, the fluorescence signals were observed with a laser scanning confocal microscope (model TCS SP8; Leica, Wetzlar, Germany). The primers used to prepare the constructs are listed in Appendix A.

### 4.9. Split Luciferase Assay

The full-length CDS of *MdbHLH093* and *MdMYB116* (without their respective stop codons) were inserted into pCB1300-Cluc and pCB1300-Nluc vectors, respectively. The plasmids were transformed into *Agrobacterium* Gv3101 and then penetrated into 4-week-old tobacco leaves. After 2 days of infiltration, the firefly luciferase substrate (0.3 mg/mL) was evenly applied to the backs of leaves, which were then placed in the dark for 10 min. A charge-coupled device camera (iKonM 934, Andor Technology, Belfast, Northern Ireland) and Winview 32 software were used for luciferase imaging. pCB1300-*MdbHLH093*-Cluc+pCB1300-Nluc, pCB1300-*MdMYB116*-Nluc+pCB1300-Cluc, and pCB1300-Cluc+pCB1300-Nluc served as the controls.

### 4.10. Gene Expression Analysis by RT-qPCR

Total RNA was extracted from *A. thaliana* and apples with the E.Z.N.A. plant RNA kit (Omega Bio-tek, Norcross, GA, USA). The Hifair III 1st Strand cDNA Synthesis SuperMix kit for qPCR (gDNA digester plus) (Yeasen, Shanghai, China) was used for reverse transcription, and operated according to the manufacturer’s instructions. Thereafter, the synthesized cDNA was diluted six-fold to serve as the qPCR template, and fluorescence quantification was performed using the Taq SYBR Green qPCR Premix kit (universal) (Yugong Biotech, Nanjing, China) in a 20 µL reaction system. The cycling conditions were as follows: 95 °C for 30 s, followed by 40 cycles of 95 °C for 5 s and 60 °C for 30 s. Relative expression was analyzed by the 2^−∆∆Ct^ method [46]. SigmaPlot 12.0 (Systat Software, San Jose, CA, USA) was used to generate the graphs.

### 4.11. Statistical Analysis

Statistical analysis was conducted using Student’s two-tailed *t* test (* *p* < 0.05, ** *p* < 0.01). Data were generated from three biological repeats. Error bars indicate standard error of the mean.

## 5. Conclusions

The transcription factor *MdbHLH093*, which is localized in the nucleus, has transcriptional activation activity. MdbHLH093 can positively regulate the resistance of apples to powdery mildew by regulating the accumulation of H_2_O_2_ and the expression of genes related to the SA signaling pathway, and it can interact with the transcription factor of the resistance gene *MdMYB116* to control plant disease resistance. Therefore, this study provides important information on the role of *MdbHLH093* in powdery mildew resistance.

## Figures and Tables

**Figure 1 ijms-24-09390-f001:**
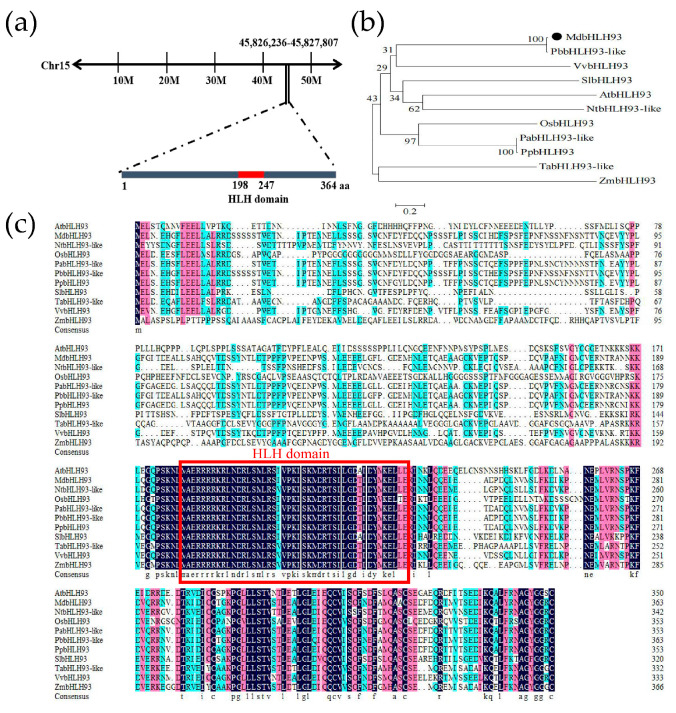
Sequence analysis of the MdbHLH093 isolated from the apple cultivar Gala. (**a**) Chromosomal location of *MdbHLH093*. (**b**) Phylogenetic analysis of MdbHLH093 and its homologs in 10 different species. The conserved HLH domain is outlined in red. The accession numbers of the bHLHs are as follows: *Arabidopsis thaliana*, AtbHLH93 (NP_569014.1); *Nicotiana tabacum*, NtbHLH93-like (XP_016490472.1); *Oryza sativa*, OsbHLH93 (XP_015629094.1); *Prunus avium*, PabHLH93-like (XP_021832105.1); *Pyrus bretschneideri*, PbbHLH93-like (XP_009349682.1); *Prunus persica*, PpbHLH93 (XP_007226892.2); *Solanum lycopersicum*, SlbHLH93 (XP_004233470.1); *Triticum aestivum*, TabHLH93-like (XP_044425383.1); *Vitis vinifera*, VvbHLH93 (XP_002285595.1); and *Zea mays*, ZmbHLH93 (AQK67253.1). (**c**) Multiple sequence alignment of MdbHLH093 and its homologs. Black means 100 percent homology, red means more than 75 percent homology, and blue means more than 50 percent homology.

**Figure 2 ijms-24-09390-f002:**
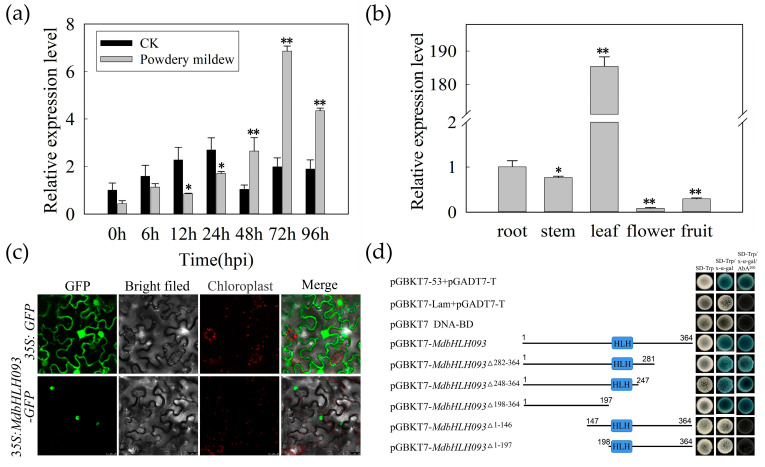
Expression pattern analysis of *MdbHLH093*. (**a**) Quantitative real-time PCR (qRT–PCR) analysis of *MdbHLH093* in plants infected with powdery mildew. (**b**) Relative expression of *MdbHLH093* in apple root, stem, leaf, flower and fruit. *MdTubulin* was used as an internal reference gene. Error bars are calculated from three biological experiments and show the standard deviation of the mean. Asterisks indicate statistical significance (*, *p* < 0.05; **, *p* < 0.01; Student’s *t*-test). (**c**) Subcellular localization of MdbHLH093 in tobacco leaves. Green fluorescent protein (GFP) signals were detected with a laser confocal microscope. Scale bar = 25 μm. (**d**) The *MdbHLH093* Yeast autoactivation transformation experiment, co-transformation of AD/T with BD/P53 or BD/Lam into yeast cells was used as positive and negative controls, respectively.

**Figure 3 ijms-24-09390-f003:**
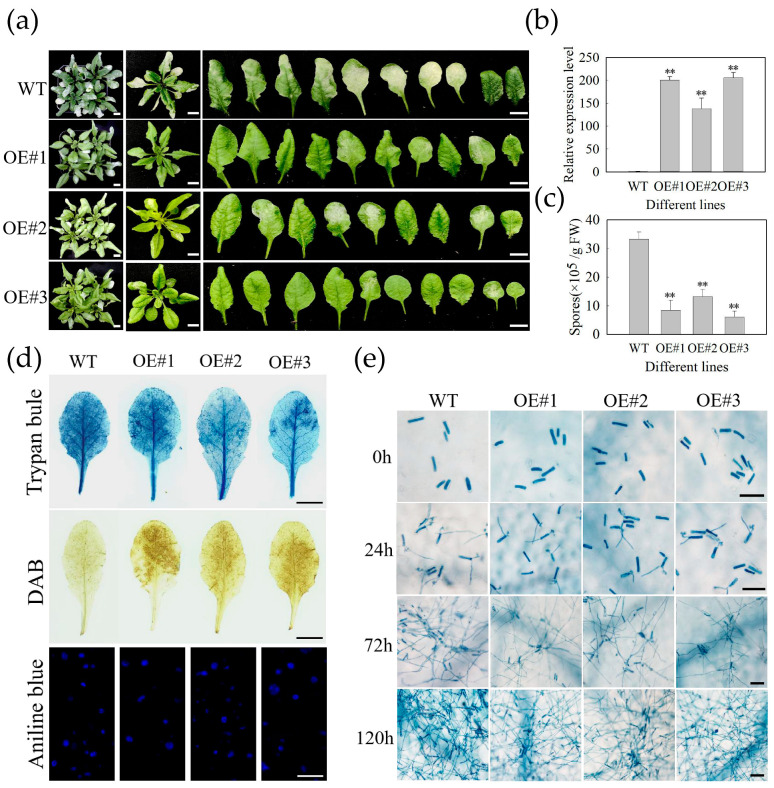
Overexpression of *MdbHLH093* in *A. thaliana* enhances resistance to powdery mildew. (**a**) Phenotypes of wild type and transgenic lines (OE # 1,2,3) after 7 days of infection with powdery mildew, Scale bar = 1 cm. (**b**) Relative gene expression of *MdbHLH093* in *A. thaliana* transgenic lines. (**c**) Number of spores per gram of fresh leaves at 7 dpi (days post-inoculation). Error bars are calculated from three biological experiments and show the standard deviation of the mean. Asterisks indicate statistical significance (**, *p* < 0.01; Student’s *t*-test). (**d**) Lesion area, H_2_O_2_, and callose accumulation after inoculation with pathogenic bacteria. Scale bar = 1 cm. (**e**) Comparison of the growth and development of powdery mildew on leaves of WT and transgenic *A. thaliana*. Scale bar = 50 μm.

**Figure 4 ijms-24-09390-f004:**
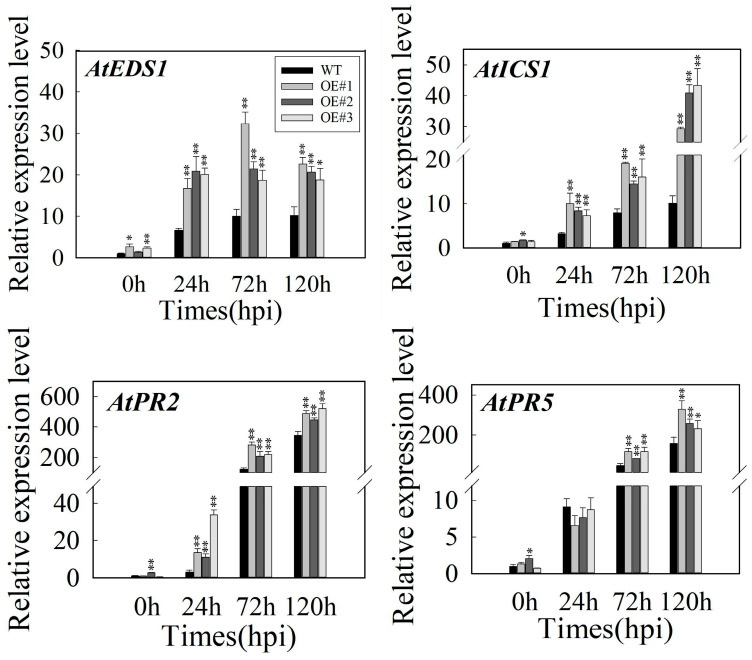
Expression analysis of disease resistance genes in transgenic *Arabidopsis*. Transcription levels of SA-pathway-related genes (*AtEDS1*, *AtICS1*, *AtPR2*, *AtPR5*) were analyzed after the 0, 24, 72, and 120 hpi infection of the *MdbHLH093* overexpression’s strain and wild type. *AtActin2* was used as an internal reference gene. Error bars are calculated from three biological experiments, and show the standard deviation of the mean. Asterisks indicate statistical significance (*, *p* < 0.05; **, *p* < 0.01; Student’s *t*-test).

**Figure 5 ijms-24-09390-f005:**
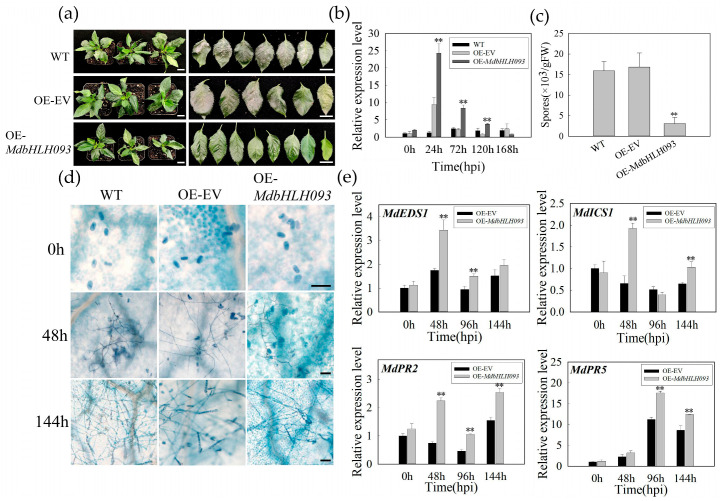
The transient overexpression of *MdbHLH093* in tissue culture plantlet Gala-3 could enhance resistance to powdery mildew. (**a**) Disease symptoms on infiltrated leaves (wild-type [WT] empty overexpression vector [OE-EV], OE-*MdbHLH093*) after *P. leucotricha* inoculation. Scale bar = 1 cm. (**b**) Analysis of *MdbHLH093* expression after transient transformation. The asterisk indicates a significant difference between OE-*MdbHLH093* and OE-EV or WT. (**c**) Number of spores per gram of fresh leaves at 7 dpi. (**d**) Comparison of growth and development of powdery mildew on leaves of WT, OE-EV, and OE-*MdbHLH093*. Scale bar = 50 μm. (**e**) Analysis of transcription levels of SA-pathway-related genes (*MdEDS1*, *MdICS1*, *MdPR2*, and *MdPR5*). Error bars are from three biological experiments, and show the standard deviation of the mean. Asterisks indicate statistical significance (**, *p* < 0.01; Student’s *t*-test).

**Figure 6 ijms-24-09390-f006:**
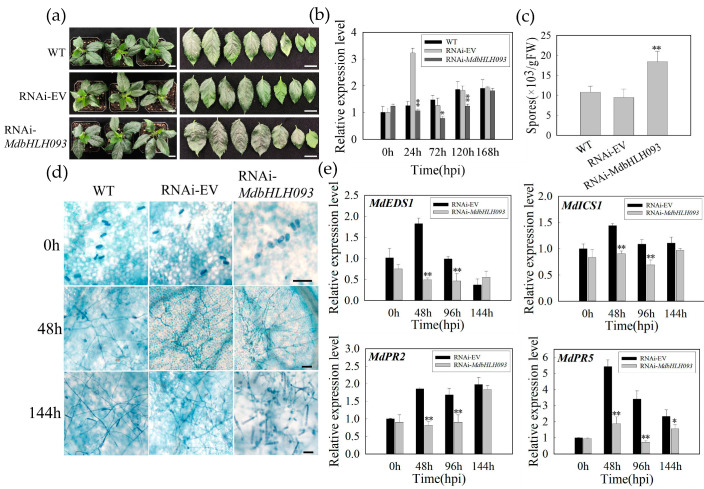
Transient silencing of *MdbHLH093* reduces resistance to apple powdery mildew. (**a**) Disease symptoms in infiltrated leaves (wild-type [WT], empty silencing vector [RNAi-EV], RNAi-*MdbHLH093*) after *P. leucotricha* inoculation. Scale bar = 1 cm. (**b**) Analysis of *MdbHLH093* expression after transient transformation. Asterisks indicate a significant difference between RNAi-*MdbHLH093* and RNAi-EV or WT. (**c**) Number of spores per gram of fresh leaves at 7 dpi. (**d**) Comparison of growth and development of powdery mildew on leaves of WT, RNAi-EV, and RNAi-*MdbHLH093*. Scale bar = 50 μm. (**e**) Analysis of transcriptional levels of SA pathway-related genes (*MdEDS1*, *MdICS1*, *MdPR2*, and *MdPR5*). Error bars are from three biological experiments and show the standard deviation of the mean. Asterisks indicate statistical significance (*, *p* < 0.05; **, *p* < 0.01; Student’s *t*-test).

**Figure 7 ijms-24-09390-f007:**
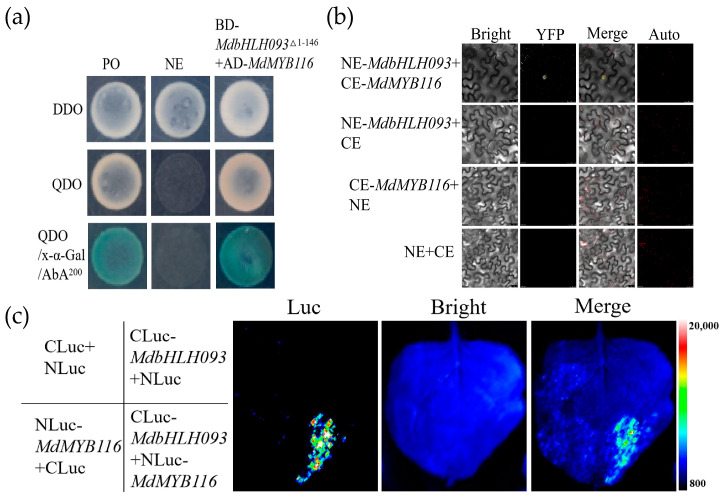
Verification of the interaction between MdbHLH093 and MdMYB116. (**a**) Yeast two-hybrid assay. pGBKT7-*MdbHLH093*^Δ1−146^ and pGADT7-*MdMYB116* plasmids were co-transformed into yeast two-hybrid gold cells. pGBKT7-53 + pGADT7-T and pGBKT7-Lam + pGADT7-T served as positive (PO) and negative controls (NE), respectively. (**b**) Bimolecular fluorescence complementation assay. Merged fluorescent and visible light images. Scale bar = 25 μm. (**c**) Interaction between MdbHLH093 and MdMYB116 was verified by split-luciferase assay. CLuc/NLuc, CLuc-*MdbHLH093*/NLuc and CLuc/NLuc-*MdMYB116* served as controls.

**Figure 8 ijms-24-09390-f008:**
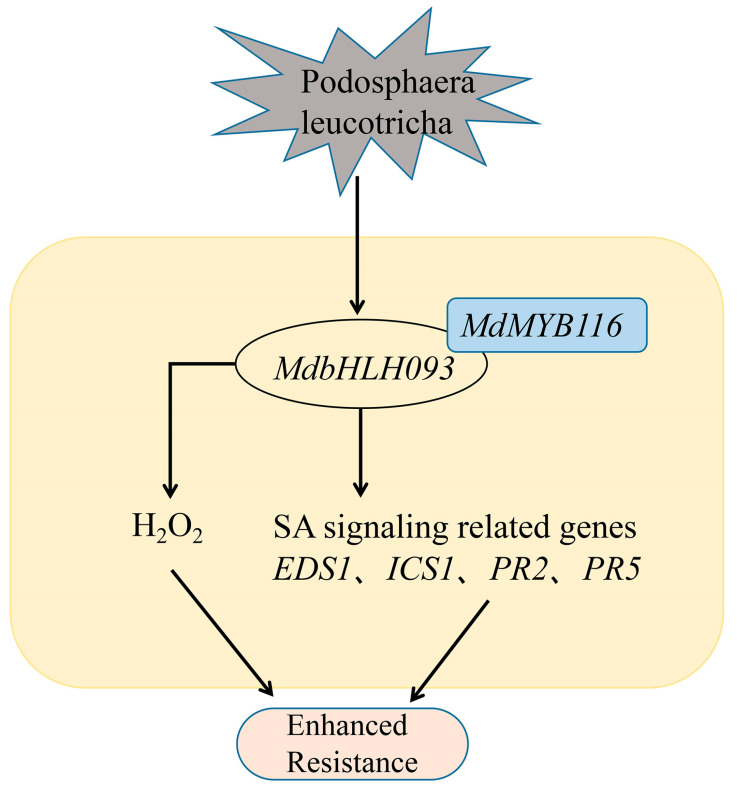
Hypothetical model depicting how *MdbHLH093* functions as a positive regulator of responses to powdery mildew infection. The transcription factor *MdbHLH093* can interact with *MdMYB116* to enhance resistance to *P. leucotricha* through hydrogen peroxide (H_2_O_2_) and salicylic acid (SA) signaling pathways, as well as improves the transcription levels of defense-related genes. Arrows represent positive effects.

## Data Availability

The data supporting the findings of this study are available within the article and its Appendix A.

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
