# Peer review of "Transcription Factor MdbHLH093 Enhances Powdery Mildew Resistance by Promoting Salicylic Acid Signaling and Hydrogen Peroxide Accumulation"

_ijms, 2023, doi:10.3390/ijms24119390_

Round 1
Reviewer 1 Report (New Reviewer)
Authors Hai Ma et al. describe a very interesting effect of MdbHLH093 transcription factor on plant defense mechanisms during the infection of apple with powdery mildew. The manuscript is globally well written.
I've found only some minor revisions:
#1. How authors explain the differences between the wt and RNAi-EV after 24 hours in figure 6b (analysis of MdbHLH093 expression after transient transformation)? Authors did not mention anything in the description of the results.
#2. The 24th reference was not founded for reading. Please check the accuracy of this reference and all the others reported in the manuscript.
Author Response
Please see the attachment.

Reviewer 2 Report (New Reviewer)
The work of Hai Ma et al. presents a significant amount of data demonstrating the role of a bHLH transctiption factor (MdbHLH093) in regulating powdery mildew resistance by promoting salicylic acid signaling and hydrogen peroxide accumulation in the apple tree. The work is solid and the conclusions derived are appropriate.
I have not any comments or suggestions and the article can be published as it is.
Author Response
Please see the attachment.

This manuscript is a resubmission of an earlier submission. The following is a list of the peer review reports and author responses from that submission.
Round 1
Reviewer 1 Report
This study describes the results of a reverse genetics approach to identifying the role of a apple bHLH transcription factor in promoting resistance to powdery mildew via SA signaling pathways. They use overexpression and RNAi knockdowns of the tomato bHLH093 TF in Arabidopsis and apple to show improved or decreases resistance to powdery mildew, and make headway into mechanisms by examining defense responses including H2O2 production, callose deposition, and SA responsive gene expression.
Generally, the results are presented clearly and interpreted accurately. They study design and methods are described adequately and appear to have been conducted properly with the needed controls. I have a few suggestions for improved clarity and more careful interpretation. Some editing for English usage would be helpful as there is some awkward phrasing throughout the manuscript.
Line 84 – MdbHLH093 was identified as a candidate because it was differentially expressed in a previous study. I assume it showed increased expression in apple leaves exposed to P. leucotricha, but please clarify what tissue and whether it was up and by how much.
Please provide an explanation for the nomenclature of the bHLH093-related genes across species. Are these TFs named based on orthologs in Arabidopsis? So that AtbHLH93 is the closest related Arabidopsis gene to MdbHLH093 (and similarly, tobacco bHLH93 is closest in that species)? Ideally, this would be validated in phylogenetic analyses if this has not already been published looking at the entire gene family between apple, Arabidopsis, and whatever other relevant species should be examined (tobacco, other 10 species included in your Fig. phylogeny). Inclusion of a gene family phylogenetic analysis in this paper, either as supplemental or in place of Figure 1 would be beneficial to demonstrate that MdBHLH093 and AtbHLH93 (and bHLH93’s from all the other species) are the closest orthologs.
Some of the experiments done using stains and microscopy need more careful interpretation. Line 180 and Line 363 says H2O2 was “significantly higher” in transgenic plants, but I do not see that statistics were run. You can say “visibly higher”, but not significantly. Line 369 says no significant change in callose content. You can say “no obvious change” but you did not test significance that I see.
In figures 5 and 6, the images (panel d) do not show clearly that pathogen growth or spore production was inhibited. Perhaps better images or making it larger or somehow highlighting what the differences are would help. Or find another way to show inhibition of pathogen growth if the images do not show it clearly.
In the Y2H assays shown in Fig. 6, the deletion derivative of bHLH093 missing the N-terminus is used. Did you also try the full length protein? Also, isn’t a negative control with only the MYB116 and no bHLH needed?
I recommend that all figures be made larger. A lot of the text embedded in the figures is too small to read. Images would be more informative if larger also.
Some of the cited literature in the introduction and discussion is not relevant to this story. bHLH TFs do all kinds of things and you do not need to summarize work on bHLHs from other species that are not related to pathogen infection, in my opinion.
Figure 1a – Expression is in leaf tissue?
Line 58 – bHLH cannot “control” drought stress, perhaps “can promote tolerance to..”
Line 186- “plants” should be “the pathogen” I believe.
Reviewer 2 Report
Good job and an informative manuscript.
To improve the manuscript, please add some information:
1. In materials and methods 4.2: Add information the volume of spore suspension
2. M&M 4.3: Please add method used for primer designing for this study (relates to Table S2)
3. Relates to Fig. 3 Please add in the M&M : The method to count number spores per gram. Did you dissolve the leaves or how did you make it.
4. The important thing in discussion, relates to your propose mechanism in Figure 8. How did you conclude that the role of H2O2 and SA-related genes were parallel in mechanism of MdbHLH093 functions as a positive regulator of responses to powdery mildew? How did you conclude that H2O2 was not in the upstream of increasing the SA-related genes?
Or whether it needs further research to confirm on the position of MdbHLH093, H2O2 and SA-related genes?
Please explain about this and state in your discussions.
Reviewer 3 Report
The current manuscript is a good work dealing with “Transcription factor MdbHLH093 enhances powdery mildew resistance by promoting salicylic acid signaling and hydrogen peroxide accumulation”
The authors are responsible for the following comments.
-General comments: In order to improve the manuscript, I encourage the authors to correct the English edition.
-Put the gene and plant scientific names in italic.
-Put the full name for any initials when mentioned the first time.
For the supplementary tables:
Please delete the S1 table, no need for it
Move the S2 table to the M&M section and add the accession number for the studied genes to the table data, at the same table, adjust the format of letter type for both forward and reverse primers.
The abstract:
- In line 18, replace (The MdbHLH093 gene was significantly induced…..) with (The expression of MdbHLH093 gene was significantly induced….).
- Introduction: line 97, " In further studies, we observed that MdbHLH093 physically.." which study, I understood that this paragraph explores the aim of the current study, please clarify what do you mean?
- Results: line 105, replace "1095 base pairs" with "1095 bp"
- In line 112, replace "software, and all genes contained an HLH.." with " software, all genes contained an HLH…"
- line 131, it will be better if the authors used this expression "To analyze the spatial regulation where MdbHLH093 may play a major role…." instead of "To analyze the tissues where MdbHLH093 may play a major role…"
- In line 134, replace "more than 100 times…" with " more than 100 fold …"
- Lines 175 and 176, replace "lower than on those of the wild type.." with " lower than those on the wild type leaves…"
- Line 185, A. thaliana
- Line 258, P. leucotricha, please write the full name for the first time
- Line 268, replace "and detected the expression level of MdbHLH093." With "and the expression level of MdbHLH093 was detected."
-The M&M: at the plant materials and growth condition section the authors have to mention the name of the apple genotype under study.
